# Consistent Solution Strategy for Static Equilibrium Workspace and Trajectory Planning of Under-Constrained Cable-Driven Parallel and Planar Hybrid Robots

Qingjuan Duan *, Quanli Zhao and Tianle Wang

School of Mechano-Electronic Engineering, Xidian University, Xi'an 710071, China
* Correspondence: qjduan@xidian.edu.cn

**Abstract:** This paper presents a consistent solution strategy for static equilibrium workspaces of different types of under-constrained robots. Considering the constraint conditions of cable force and taking the least squares error of the static equilibrium equation as the objective, the convex optimization solution is carried out, and the static equilibrium working space of the under-constrained system is obtained. A consistent solution strategy is applied to solve the static equilibrium workspaces of the cable-driven parallel and planar hybrid robots. The dynamic models are presented and introducing parameters that are applied to make the system stable for point-to-point movements. Based on this model, the traditional polynomial-based point-to-point trajectory planning algorithm is improved by adding unconstrained parameters to the kinematic law function. The constraints of the dynamics model are incorporated into the trajectory planning process to achieve point-to-point trajectory planning for the under-constrained cable-driven robots. Finally, under-constrained cable-driven parallel robots with three cables and planar hybrid robot with two cables are taken as examples to carry out numerical simulation. The final results show that the point-to-point trajectory planning algorithm introducing parameters is effective and feasible and can provide theoretical guidance for the design of subsequent under-constrained robots.

**Keywords:** static equilibrium workspace; under-constrained cable-driven robot; consistent solution strategy; trajectory planning



## 1. Introduction

The cable-driven robot is a mechanism that employs cables in place of rigid-body to control the end-effector pose.

The classification of cable-driven parallel robots (CDPRs) was introduced by Ming and Higuchi [1]. A cable-driven robot with $n$ degrees-of-freedom and $m$ cables can be classified into the following four categories: (1) $n + 1 < m$: These robots are referred to as redundantly restrained positioning mechanisms (RRPM), the static forces of the robot are generally undefined. (2) $n + 1 = m$: These robots are called completely restrained positioning mechanisms (CRPM). All degrees of freedom can be controlled through cables. (3) $m = n$: This type of robot is called incompletely restrained positioning mechanism (IRPM). When external forces such as gravity applied, the robot is fully constrained. It can withstand a limited range of wrenches. (4) $m < n$: The robot is under-constrained positioning mechanism (URPM) and in general cannot withstand arbitrary external forces and torques. Due to the under-constrained nature, these robots have one feasible solution for cable tensions and mostly works under gravity conditions. This classification method is also applicable to cable-driven planar hybrid robots.

According to the structure of cable-driven robot, it is generally divided into series mechanism and parallel mechanism.

Cable-driven robot has the characteristics of simple structure, flexibility, large workspace, low inertia, high load rate, etc. It has a wide range of applications, such as: Five-hundred-meter

Aperture Spherical radio Telescope (FAST), wind tunnel test, rehabilitation training, sports photography, etc. [2–5], it has become a hot spot in robotics research recent years.

The under-constrained cable-driven robots (UCR), with few drivers and low cost, has its special purpose, attracting more and more scholars' research interest. Carricato et al. [6–8] from Italy studied the cable-driven parallel robots with less than six cables, provided a geometrico-static model, and assessed the stability of static equilibrium within the framework of a constrained optimization problem. Several examples are provided, concerning robots with a number of cables that range from 2 to 4. Berti et al. [9] proposed a method based on interval analysis to solve the positive geometric statics problem of an under-constrained cable-driven parallel robot, and find all possible equilibrium poses of the end effector under a given cable length. Liu Xin et al. [10] proposed a consistent algorithm for solving the workspace of a cable-driven parallel robot under different constraints. Fu Ying et al. [11] conducted a dynamic analysis on the cable-driven system with four cables and six degrees of freedom. Zhao Zhigang et al. [12] proposed a comprehensive algorithm by combining the least squares method and the Monte Carlo algorithm to solve the statically balanced workspace for the cable-driven system with multi-robots. Peng Y et al. [13] analyzed the reachable workspace for spatial 3-cable under-constrained suspended cable driven parallel robots. The above-mentioned literature put forwards the solution method of static equilibrium workspace for under-constrained parallel robots. Based on this, a consistent solution strategy for the static equilibrium workspace of both cable-driven parallel & planar hybrid robots is put forward in this paper.

Ida E et al. [14] proposed a rest-to-rest trajectory planning for underactuated cable-driven parallel robots. Barbazza L et al. [15] design and optimally control an underactuated cable-driven micro–macro robot. Shang Weiwei et al. [16] proposed a geometrical approach to plan trajectories that extend beyond the static equilibrium workspace (SEW) of the mechanism. Zi B et al. [17] studied an algebraic method-based point to point trajectory planning of an under constrained cable suspended parallel robot with variable angle and height cable mast. Shi P et al. [18] studied Dimensional synthesis of a gait rehabilitation cable-suspended robot on minimum 2-norm tensions. The above-mentioned literature put forwards the planning trajectories for under-constrained parallel robots. Based on this, a consistent solution strategies of point-to-point trajectory planning introducing parameters for both cable-driven parallel & planar hybrid robots is put forward in this paper.

Firstly, the static equilibrium equations of the under constrained parallel and planar hybrid robots are established. Then, the mathematical description of the static equilibrium workspace of the under-constrained cable-driven robots (UCR) that meets the constraints of the driving motor power and cable strength is given. The characteristics of the static equilibrium equation and dynamics equation are analyzed, and the consistent solution strategy for the static equilibrium workspace and point-to-point trajectory planning introducing parameters of different types of under-constrained robots are given.

The paper is organized as follows: Section 2 presents the model for under-constrained parallel robots. Section 3 provides the model of under-constraint cable-driven planar hybrid robot. Section 4 describes the consistent solution strategy for static equilibrium workspace. Section 5 illustrates some results of static equilibrium workspace. Section 6 puts forward the consistent solution strategies for point-to-point trajectory planning. Section 7 illustrates the simulation results of trajectory planning. Finally, Section 8 draws conclusions.

## 2. Model of Under-Constrained Cable-Driven Parallel Robots

The schematic structure of a cable-driven robot with $n$ degrees-of-freedom and $m$ cables is shown in Figure 1. Here $m < n$, it's an under-constrained CDPR. The moving platform is connected to the base through m cables, the *ith* cable ($i = 1,2, \ldots ,m$) exits from the fixed base at point $A_i$, connected to the moving platform at point $B_i$. the cable length is $L_i$. $Oxyz$ is a Cartesian coordinate which fixed to the base, and $O'x'y'z'$ is the Cartesian coordinate fixed to the moving platform.

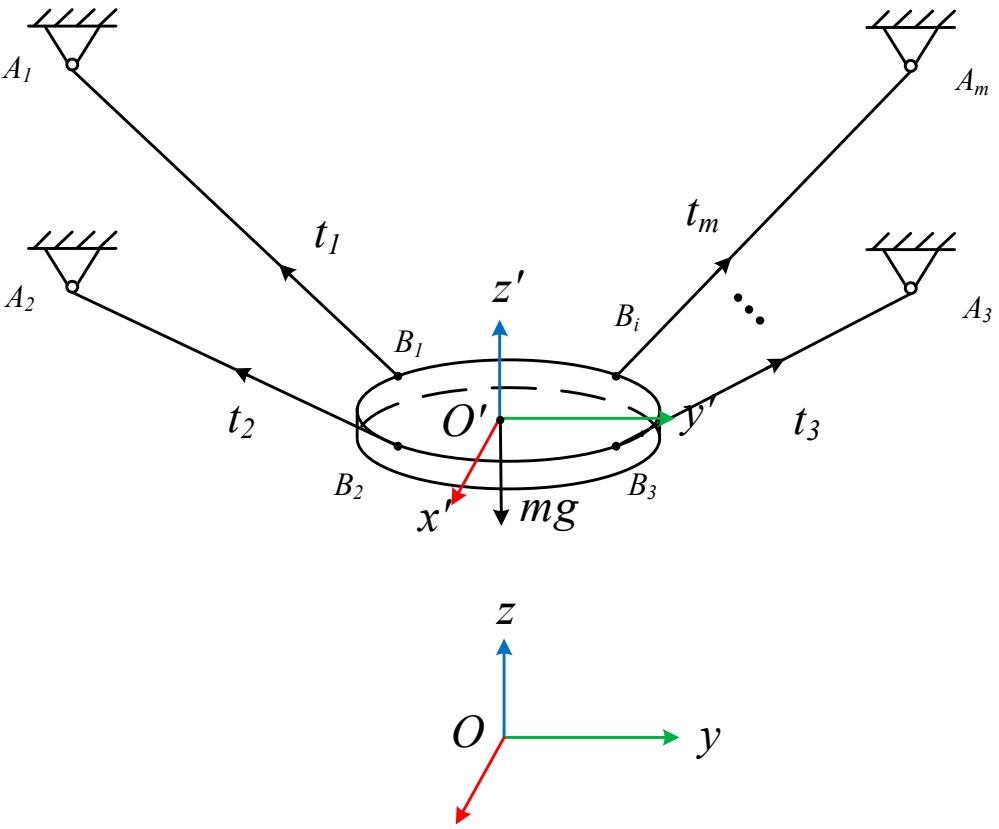

**Figure 1.** The schematic structure of a cable-driven parallel robot (If $m < n$, it is an under-constrained CDPR.).

$^{O}P_{O'} = [^{O}x_{O'} \ ^{O}y_{O'} \ ^{O}z_{O'}]^{\mathrm{T}}$ is the centroid of end effector in the $Oxyz$ frame. $^{O'}B_i = [^{O'}x_{Bi} \ ^{O'}y_{Bi} \ ^{O'}z_{Bi}]^{\mathrm{T}} (i = 1, 2, \cdots, m)$ is the vector connecting point $O'$ to the point $B_i$ in the $O'x'y'z'$ frame. $^{O}A_i = [^{O}x_{Ai} \ ^{O}y_{Ai} \ ^{O}z_{Ai}]^{\mathrm{T}} (i = 1, 2, \cdots, m)$ is the fixed base at point $A_i$, in the $Oxyz$ frame.

$^{O}R_{O'}$ represents the rotational matrix from frame $O'x'y'z'$ to frame $Oxyz$, in which $\alpha, \beta, \gamma$ are $x$-$y$-$z$ the Euler angles.

$$^{O}R_{O'} = \mathrm{rot}(x, \alpha)\mathrm{rot}(y, \beta)\mathrm{rot}(z, \gamma) = \begin{bmatrix} c\beta c\gamma & -c\beta s\gamma & s\beta \\ c\alpha s\gamma + s\alpha s\beta c\gamma & c\alpha c\gamma - s\alpha s\beta s\gamma & -c\beta s\alpha \\ s\alpha s\gamma - c\alpha s\beta c\gamma & s\alpha c\gamma + c\alpha s\beta s\gamma & c\alpha c\beta \end{bmatrix} \tag{1}$$

where $c$ represents cos, $s$ represents sin.

$^{O}B_i$ is the vector in the $Oxyz$ frame.

$$^{O}B_i = {}^{O}R_{O'}{}^{O'}B_i + {}^{O}P_{O'} \tag{2}$$

$^{O}L_i$ is the vector connecting point $B_i$ to point $A_i$ in the $Oxyz$ frame. $e_i$ is the unit vector of $^{O}L_i$.

$$e_i = \frac{^{O}L_i}{\|^{O}L_i\|_2} = \frac{^{O}A_i - {}^{O}B_i}{\|^{O}A_i - {}^{O}B_i\|_2} \tag{3}$$

Thus, the dynamics equations for end effector are shown as follows:

$$JT + G = \begin{bmatrix} m\ddot{x} \\ I\dot{\omega} + \omega \times I\omega \end{bmatrix} \tag{4}$$

where $T = [t_1 \ t_2 \ \cdots \ t_m]^T$, $t_i(i = 1, 2, \cdots, m)$ is cable tensions act on the end effector. $J = [\hat{f}_1 \ \hat{f}_2 \ \cdots \ \hat{f}_m]$ is the construction matrix, $\hat{f}_i = [e_i \quad {}^OB_i \times e_i]^T$, and $G = [0 \ 0 - mg \ 0 \ 0]^T$. $\ddot{x} = [{}^O\ddot{x}_{O'}, {}^O\ddot{y}_{O'}, {}^O\ddot{z}_{O'}]^T$ is the acceleration of the end-effector centroid $O\prime$ in the world coordinate system, and $I = {}^OR_{O'}I_{O'}{}^OR_{O'}$, $I_{O'}$ is the inertia tensor of the end-effector in the local coordinate system.

$\omega$ and $\dot{\omega}$ are the angular velocity and angular acceleration of the end-effector. $\omega$, $\dot{\omega}$ and $\varepsilon$, the Euler angle, satisfy the following relationship:

$$\omega = H(\varepsilon)\dot{\varepsilon} = \begin{bmatrix} 1 & 0 & s\beta \\ 0 & c\alpha & -s\alpha c\beta \\ 0 & s\alpha & c\alpha c\beta \end{bmatrix} \begin{bmatrix} \dot{\alpha} \\ \dot{\beta} \\ \dot{\gamma} \end{bmatrix} \tag{5}$$

$$\dot{\omega} = \dot{H}\dot{\varepsilon} + H\ddot{\varepsilon} \tag{6}$$

Substituting Equations (5) and (6) into Equation (4) yields a general expression for the system dynamics equation [19], i.e.,

$$M(q)\ddot{q} - s(q,\dot{q}) - J(q)T = 0$$
$$M(q) = \begin{bmatrix} mE_3 & 0 \\ 0 & IH \end{bmatrix}, s(q,\dot{q}) = G - \begin{bmatrix} 0 \\ I\dot{H}\dot{\varepsilon} + \omega \times I\omega \end{bmatrix} \tag{7}$$

In Equation (7), $\ddot{q} = [\ddot{x} \ \ddot{\varepsilon}]^T$ is the end-effector acceleration. $M$ is the mass matrix of the end-effector. $s$ is the force vector for a collection of Coriolis-type forces, gravitational forces, and external loads, etc. $J(q)T$ is the cable-tension to which the end-effector is subjected.

Additionally, the static equilibrium equation of the system can be expressed by

$$JT + G = 0 \tag{8}$$

## 3. Model of Under-Constrained Cable-Driven Planar Hybrid Robot

The schematic structure of a cable-driven planar hybrid robot with planar $n$-link serial robot with $n$ degrees-of-freedom and $m$ cables is shown in Figure 2. All links are connected by revolute joints to form a planar multi-link mechanism. This definition can also be generalized to space model. Here, $m < n$, it's an under-constrained cable-driven planar hybrid robot. $\{0\}$ is the base frame, $\{i\}$ is the link frame $x_iy_i$, $i \in \{1, 2, \cdots, n\}$. $\theta_i(i = 1, 2, \cdots, n)$ is the angle between link frame $\{i-1\}$ and $\{i\}$. $[\theta_1 \ \theta_2 \ \cdots \ \theta_n]^T$ is the Joint coordinates of the system.

${}^0C_i$ is the position vector of the $ith$ link centroid in the fixed frame $\{0\}$. It can be expressed by

$${}^0C_i = [{}^0x_{C_i} \ {}^0y_{C_i}]^T = ({}^0R_1 {}^1R_2 \cdots {}^{i-1}R_i){}^iC_i \tag{9}$$

where ${}^0R_1 = \begin{bmatrix} c\theta_1 & -s\theta_1 & 0 \\ s\theta_1 & c\theta_1 & 0 \\ 0 & 0 & 1 \end{bmatrix}$, ${}^iR_{i+1} = \begin{bmatrix} c\theta_{i+1} & -s\theta_{i+1} & 0 \\ s\theta_{i+1} & c\theta_{i+1} & l_i \\ 0 & 0 & 1 \end{bmatrix}$, ${}^iR_{i+1}$ represents the rotational matrix from frame $i+1$ to frame $i$, $i = 1, 2, \cdots, n-1$.

${}^0B_j^i$ is the $jth$ cable position vector of the $ith$ link in the fixed frame. It can be expressed by

$${}^0B_j^i = [{}^0x_{B_j^i} \ {}^0y_{B_j^i}]^T = ({}^0R_1 {}^1R_2 \cdots {}^{i-1}R_i){}^iB_j^i \tag{10}$$

where ${}^iB_j^i = [{}^ix_{B_j^i} \ {}^iy_{B_j^i}]^T$ is the cable position vector of the $ith$ link in the link frame $\{i\}$.

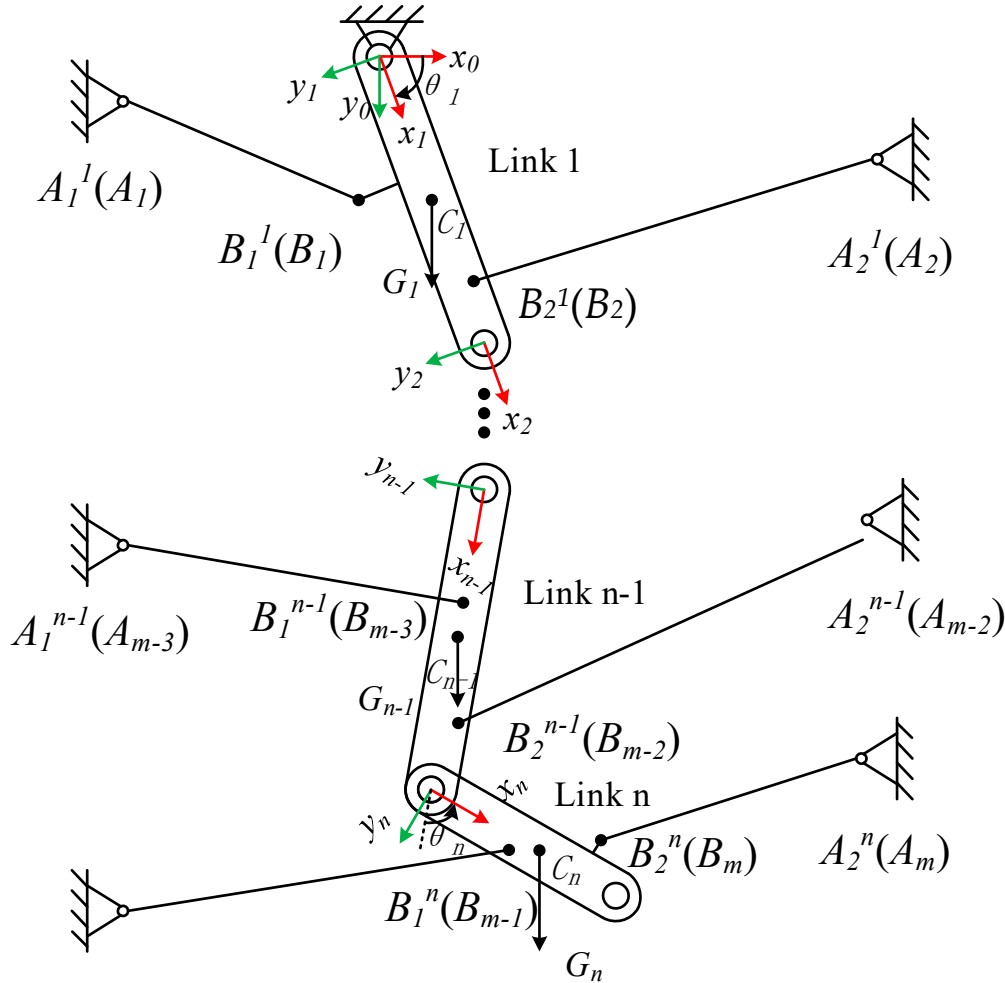

**Figure 2.** The schematic structure of a cable−driven planar hybrid robot with planar serial *n*−link with *n* degrees-of-freedom and *m* cables (If *m* < *n*, it's an under-constrained cable-driven planar hybrid robot.).

The generalized force is solved by Lagrange equation,

$$\tau = \frac{d}{d_t}\left(\frac{\partial L}{\partial \dot\theta}\right) - \frac{\partial L}{\partial \theta}$$

$$L = E_k - E_p$$

(11)

$$\tau = \frac{\partial\left(\frac{\partial L}{\partial \dot\theta}\right)}{\partial \dot\theta}\ddot\theta + \frac{\partial\left(\frac{\partial L}{\partial \dot\theta}\right)}{\partial \theta}\dot\theta - \frac{\partial L}{\partial \theta}$$

(12)

where $\tau$ is the generalized force, $L$ is the Lagrange function. $\theta = \begin{bmatrix} \theta_1 & \theta_2 & \cdots & \theta_n \end{bmatrix}^{\mathrm{T}}$ is the generalized coordinate. $E_k$ and $E_p$ are the kinetic energy and potential energy of the system, respectively.

Write Equation (12) as a generic expression,

$$M(\theta)\ddot\theta - s(\theta,\dot\theta) = \tau$$

$$M(\theta) = \frac{\partial\left(\frac{\partial L}{\partial \dot\theta}\right)}{\partial \dot\theta}, s(\theta,\dot\theta) = -\frac{\partial\left(\frac{\partial L}{\partial \dot\theta}\right)}{\partial \theta}\dot\theta + \frac{\partial L}{\partial \theta}$$

(13)

where $M(\theta)$ is the inertia matrix, and $s(\theta, \dot{\theta})$ is the gravity, centrifugal force, Coriolis force, etc.

The expression (14) can be obtained according to the principle of virtual work

$$\tau^{\mathrm{T}} \delta\theta = \sum_{i=1}^{m} T_i e_i^{\mathrm{T}} \delta(^0 B_i) = \sum_{i=1}^{m} T_i e_i^{\mathrm{T}} \left( \frac{\partial^0 B_i}{\partial \theta} \delta\theta \right) \tag{14}$$

where $m$ represents the number of cables, $T$ is the cable tension, $e$ is the unit vector of the cable tension, and $^0 B_i$ is the position of the $ith$ pulling point in the coordinate system.

From Equation (14), the relationship between generalized force and cable tension can be obtained

$$\tau = J_v^{\mathrm{T}} T \tag{15}$$

where

$$J_v^{\mathrm{T}} = \begin{bmatrix} e_1^{\mathrm{T}} \frac{\partial^0 B_1}{\partial \theta_1} & e_1^{\mathrm{T}} \frac{\partial^0 B_1}{\partial \theta_2} & \cdots & e_1^{\mathrm{T}} \frac{\partial^0 B_1}{\partial \theta_n} \\ e_2^{\mathrm{T}} \frac{\partial^0 B_2}{\partial \theta_1} & e_2^{\mathrm{T}} \frac{\partial^0 B_2}{\partial \theta_2} & \cdots & e_2^{\mathrm{T}} \frac{\partial^0 B_2}{\partial \theta_n} \\ \vdots & \vdots & \ddots & \vdots \\ e_m^{\mathrm{T}} \frac{\partial^0 B_m}{\partial \theta_1} & e_m^{\mathrm{T}} \frac{\partial^0 B_m}{\partial \theta_2} & \cdots & e_m^{\mathrm{T}} \frac{\partial^0 B_m}{\partial \theta_n} \end{bmatrix}^{\mathrm{T}}$$

where $J_v$ is the pseudo-Jacobian of the constraint equations, $J_v^{\mathrm{T}}$ is the $n \times m$ matrix, $T$ is the $m \times 1$ cable tension matrix.

Combining Equations (13) and (15), the system dynamics equation is obtained as follows.

$$M(\theta)\ddot{\theta} - s(\theta, \dot{\theta}) - J_v^{\mathrm{T}}(\theta)T = 0 \tag{16}$$

Since the system is in static equilibrium, and the generalized velocity $\dot{\theta} = 0$, Formula (11) can be simplified as

$$\tau = \frac{\partial E_p}{\partial \theta} \tag{17}$$

where $E_p = \sum_{i=1}^{n} (-n_i g)^0 y_{C_i}$. Taking the origin of the coordinate system as the reference point of potential energy.

Combine Equations (15) and (17), yields

$$J_v^{\mathrm{T}} T = \frac{\partial E_p}{\partial \theta} \tag{18}$$

## 4. Consistent Solution Strategy for Static Equilibrium Workspace

$\begin{bmatrix} ^O x_O, & ^O y_O, & ^O z_O, & \alpha & \beta & \gamma \end{bmatrix}^{\mathrm{T}}$ is the pose of the parallel robot, $\begin{bmatrix} \theta_1 & \theta_2 & \cdots & \theta_n \end{bmatrix}^{\mathrm{T}}$ is the Joint coordinates of the planar hybrid robot. They are uniformly written into generalized coordinates $X = \begin{bmatrix} x_1 & x_2 & \cdots & x_n \end{bmatrix}^{\mathrm{T}}$.

Combine Equations (8) and (18), yields

$$J(X)T = W(X) \tag{19}$$

where $W$ is the external force.

### 4.1. The Definition of Static Equilibrium Workspace

Due to the limitation of motor torque and cable strength, the tension of the cable must be within a certain range. Therefore, for a robot with $n$ degrees of freedom pulled by $m$ cables, the mathematical description of its static equilibrium workspace is as follows:

$$X = \begin{bmatrix} x_1 & x_2 & \cdots & x_n \end{bmatrix}^{\mathrm{T}}$$

$$\exists t_{\min} \leq t_i \leq t_{\max} (t_{\min} > 0 \cap t_{\max} > 0 \cap t_{\max} > t_{\min}, i = 1, 2, \cdots, m) \tag{20}$$

$$J(X)T = W(X)$$

where $X$ represents the generalized coordinate. $t_{\min}$, $t_{\max}$ are the minimum and the maximum allowable tension of the cable, respectively. $T = \begin{bmatrix} t_1 & t_2 & \cdots & t_m \end{bmatrix}^{\mathrm{T}}$ is the cable tension, and $X$ belongs to the Static equilibrium workspace.

### 4.2. Consistent Solution Strategy

For a cable-driven robot with $n$ degrees-of-freedom and $m$ cables, here $m < n$, it's an Under-constrained Cable-driven Robot (UCR). The Static equilibrium workspace of the under-constrained cable-driven system is analyzed as follows:

In static equilibrium Equation (19), $J$ is $n \times m$ matrix, $T$ is $m \times 1$ matrix. For the under-constrained cable-driven system, the solution of the equation is in the form of the least squares solution, and it can be expressed by $T = J^\dagger W$. When this solution is within the limit of cable force, it is considered that the pose (generalized coordinate) is the static equilibrium point satisfying Equation (20).

Then, the following inequality holds

$$\begin{aligned} \|J(X)T - W(X)\|_2 &< \sigma* \\ T = J^\dagger(X)W(X), t_{\min} &\leq t_i \leq t_{\max} \end{aligned} \tag{21}$$

where $\sigma*$ is the error of the least square solution.

To sum up, the consistent solution strategy for the Static equilibrium workspace of the UCRs is as follows:

1.  For the under-constrained robot, given the lower limit and the upper limit of cable tension $t_{\min}$ and $t_{\max}$ separately, selected the controllable degrees of freedom and uncontrollable degrees of freedom $X_a = [x_1, \ldots, x_k]$ and $X_b = [x_1, \ldots, x_{n-k}]$ separately.
2.  Judge the search range $k$ of the system dimension, set the search step $\delta_1, \ldots, \delta_k$, and generate the pose set $\mathbf{Q}$ to be searched.
3.  Take a pose $X_a{}^i$ to be searched from the set and bring it into Formula (19) to solve the Jacobian matrix $J$.
4.  Set convex optimization solution goal: $\sigma = \|J(X^i)T - W(X^i)\|_2$, nonlinear constraint: $t_{\min} \leq t_i \leq t_{\max}$. Use the interior point method to solve it.
5.  Check whether the results of step 4 meets $\sigma < \sigma^*$. If the condition is true, there is a Static equilibrium point $X^i = [X_a{}^i, X_b{}^i]$ in the controllable generalized coordinate $X_a{}^i$ that meets the cable force condition, turn to step 6, otherwise, there is no Static equilibrium point in the controllable posture.
6.  Calculate the next pose $X_a{}^{i+1}$ to be searched.

## 5. Simulation Results of Static Equilibrium Workspace

### 5.1. 3−6 Under-Constrained Parallel Robot

Solve the Static equilibrium workspace of 3−6 (3 cables pulling 6 degrees of freedom) under-constrained parallel robot. The mass of end effector is $m = 0.935$ kg. See Table 1 for specific parameters.

**Table 1.** 3−6 Configuration of cable-driven parallel robot.

| Coordinates of Anchor Point (m) | Coordinates of the Hinge Point between the Cable and the End Actuator (m) |
|---|---|
| $^{O}A_1 = \begin{bmatrix} 0.9 & 0 & 1.8 \end{bmatrix}^T$ | $^{O'}B_1 = \begin{bmatrix} 0.2265 & -0.2275 & 0 \end{bmatrix}^T$ |
| $^{O}A_2 = \begin{bmatrix} 0 & 1.8 & 1.8 \end{bmatrix}^T$ | $^{O'}B_2 = \begin{bmatrix} 0 & 0.2275 & 0 \end{bmatrix}^T$ |
| $^{O}A_3 = \begin{bmatrix} -0.9 & 0 & 1.8 \end{bmatrix}^T$ | $^{O'}B_3 = \begin{bmatrix} -0.2265 & -0.2275 & 0 \end{bmatrix}^T$ |

$t_{\min} = 0.1$ N is the lower limit of cable tension, and $t_{\max} = 10$ N the upper limit of cable tension. $\sigma* = 1 \times 10^{-8}$ is the error. The controllable degree of freedom is $X_a = \begin{bmatrix} ^{O}x_{O'}, ^{O}y_{O'}, ^{O}z_{O'} \end{bmatrix}$, and the uncontrollable degree of freedom is $X_b = [\alpha, \beta, \gamma]$, the rotation angle of $x$-axis, $y$-axis, and $z$-axis, respectively. Considering the volume of end effector and cable, set the search interval is

$$\mathbf{S}_x \in (-0.6, 0.6), \mathbf{S}_y \in (0.3, 1.6), \mathbf{S}_z \in (0.27, 1.52)$$

The search step of $\delta_x, \delta_y, \delta_z$ is 0.05m, and the pose set $\mathbf{Q}$ to be searched is generated.

Matlab is used to solve the static equilibrium workspace of 3−6 cable-driven parallel robot.

It can be seen from Figure 3 the 3D view, *xoy* planar view and *xoz* planar view of the Static equilibrium workspace that the Static equilibrium workspace of the 3−6 under-constrained parallel robot is approximately a triangular prism, whose cross section is symmetrical along the *x*-axis, and there are two narrow workspace cracks in the lower half ($z < 0.75$ m) of the triangular prism. Compared with the system schematic diagram 1, this situation occurs because when the center of mass of the end actuator is low, the tension force of the three cables cannot be balanced with gravity, so the pose of the end actuator cannot be kept stable. The anchor points $^{O}A_1$ and $^{O}A_3$ are symmetrical along the *y*-axis, and the anchor point $^{O}A_2$ is on the axis, so that the cross section of its Static equilibrium workspace is symmetrical about the *y*-axis.

*5.2. 0−0−2 Under-Constrained Planar Hybrid Robot*

Solve the Static equilibrium workspace of the 0−0−2 under-constrained planar hybrid robot (no cable attached on the first and second links, and two cables attach on the third link). The Tables 2 and 3 show the specific parameters of 0−0−2 cable-driven planar hybrid robot system.

**Table 2.** 0−0−2 configuration of cable-driven planar hybrid robot.

| Coordinates of Anchor Point (m) | Coordinates of the Hinge Point on the End Effector (m) |
|---|---|
| $^{O}A_1 = \begin{bmatrix} -0.9 & 0.6 \end{bmatrix}^T$ | $^{3}B_1 = \begin{bmatrix} 0.09 & 0.08 \end{bmatrix}^T$ |
| $^{O}A_2 = \begin{bmatrix} 0.9 & 0.5 \end{bmatrix}^T$ | $^{3}B_2 = \begin{bmatrix} 0.16 & -0.03 \end{bmatrix}^T$ |

**Table 3.** Link parameters.

| The Quality of the Link(kg) | The Length of the Link(m) |
|---|---|
| $m_1 = 11.8$ | $l_1 = 0.45$ |
| $m_2 = 4.5$ | $l_2 = 0.4$ |
| $m_3 = 1.1$ | $l_3 = 0.25$ |

The lower limit of the cable tension is $t_{\min} = 5$ N, the upper limit of the cable tension is $t_{\max} = 200$ N, $\sigma* = 1 \times 10^{-6}$ is the error.

$X_a = [\theta_1, \theta_2]$ is the controllable degrees of freedom, and $X_b = [\theta_3]$ is the uncontrollable degrees of freedom.

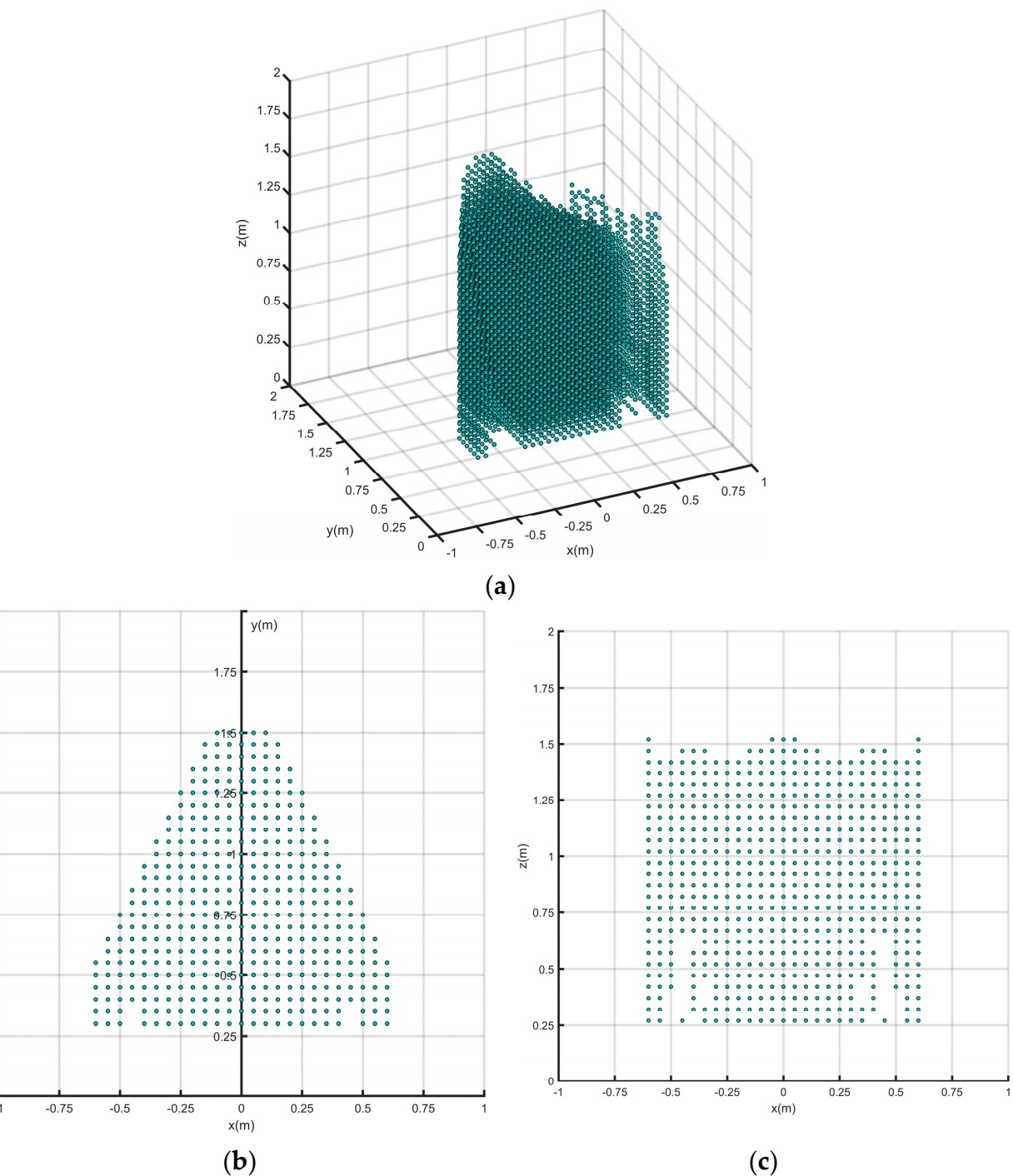

**Figure 3.** 3−6 Static equilibrium workspace of cable-driven parallel robot. (**a**) 3D Static Equilibrium Workspace; (**b**) *xoy* planar view of Static Equilibrium Workspace; (**c**) *xoz* planar view of Static Equilibrium Workspace.

Set the search interval is $\mathbf{S}_{\theta_1} \in (65°, 95°) \& \mathbf{S}_{\theta_2} \in (-40°, 0°)$.

The search step $\delta_{\theta_1}, \delta_{\theta_2}$ is 1°, and the pose set to be searched is **Q**.

The static equilibrium working space of the 0−0−2 cable-driven robot is solved by MATLAB programming.

According to Figure 4, the static equilibrium workspace of the 0−0−2 under-constrained cable-driven planar hybrid robot is a curved surface. From Figure 4b, $\theta_1 \theta_2$ planar view of the Static equilibrium workspace, the Static equilibrium points in the pose set are mainly

distributed in the lower right corner area. The uncontrollable degrees of freedom of the static equilibrium point belongs to $\theta_3 \in (-80°, -20°)$ from Figure 4c,d.

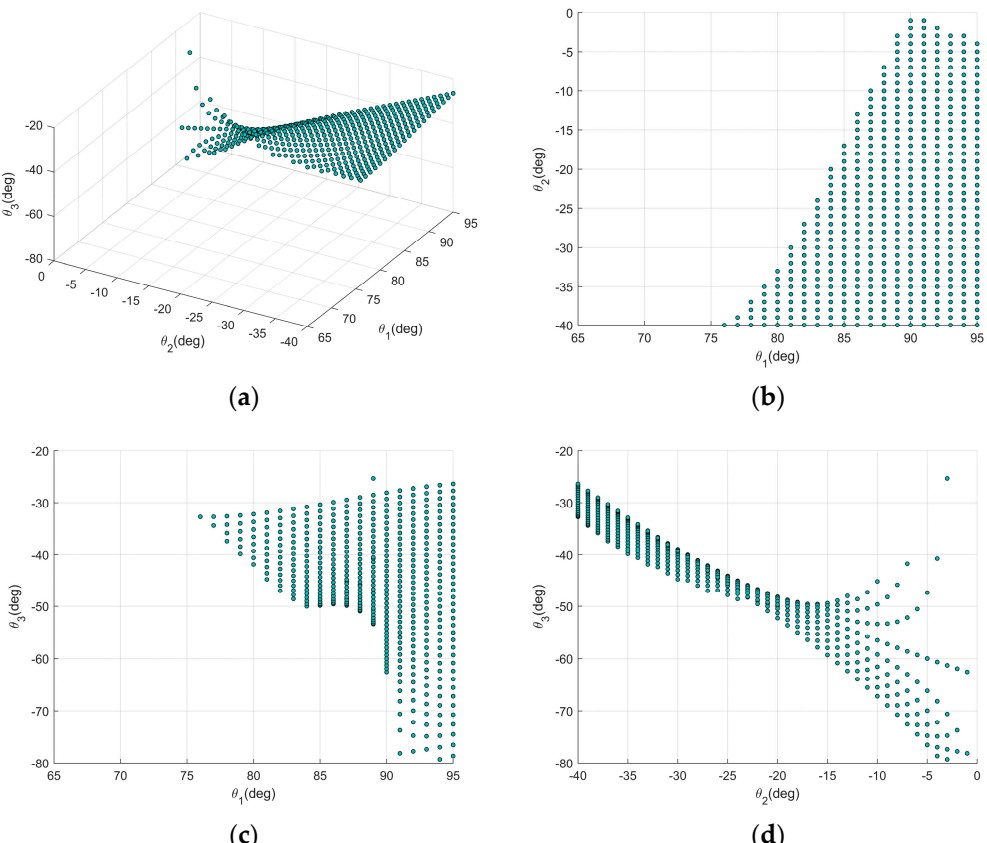

**Figure 4.** Static Equilibrium Workspace of $0-0-2$ unconstrained cable-driven planar hybrid robot. (**a**) 3D Static Equilibrium Workspace; (**b**) $\theta_1\theta_2$ planar view of Static Equilibrium Workspace; (**c**) $\theta_1\theta_3$ planar view of Static Equilibrium Workspace; (**d**) $\theta_2\theta_3$ planar view of Static Equilibrium Workspace.

## 6. Consistent Solution Strategies for Point-to-Point Trajectory Planning

### 6.1. Analysis of the Dynamics of Under-Constrained Systems

The end poses $\begin{bmatrix} ^O x_O, & ^O y_O, & ^O z_O, & \alpha & \beta & \gamma \end{bmatrix}^{\mathrm{T}}$ of the parallel robot and Joint coordinates $\begin{bmatrix} \theta_1 & \theta_2 & \cdots & \theta_n \end{bmatrix}^{\mathrm{T}}$ of the planar hybrid robot are written in generalized coordinates $X = \begin{bmatrix} x_1 & x_2 & \cdots & x_n \end{bmatrix}^{\mathrm{T}}$.

Write Equations (7) and (16) uniformly as:

$$M(X)\ddot{X} - s(X, \dot{X}) - J(X)T = 0 \tag{22}$$

where $\ddot{X}$ is the acceleration in generalized coordinates, $M(X)$ is the mass matrix, and $J(X)T$ is the vector of the cable tensions.

For the under-constrained system, the controllable degrees of freedom, $X_a$, are $m$ dimensional vectors. The uncontrollable degrees of freedom, $X_u$, are $n - m$ dimensional vectors. As a result, Equation (22) can be written in the form of

$$\begin{bmatrix} M_{aa} & M_{au} \\ M_{ua} & M_{uu} \end{bmatrix} \begin{bmatrix} \ddot{X}_a \\ \ddot{X}_u \end{bmatrix} - \begin{bmatrix} s_a \\ s_u \end{bmatrix} - \begin{bmatrix} J_a \\ J_u \end{bmatrix} T = 0 \tag{23}$$

For Equation (23), when $X, \dot{X}, \ddot{X}_a$ is known, the acceleration $\ddot{X}_u$ and the cable force $T$ for the system at this moment can be obtained.

By presenting the acceleration terms for the (UDFS) uncontrollable degrees of freedom, we obtain the expressions for $\ddot{X}_u$

$$\ddot{X}_u = M_{uu}^{-1}\left(s_u + J_u T - M_{ua}\ddot{X}_a\right) \tag{24}$$

Substituting the expression for $\ddot{X}_u$ into Equation (23), we obtain the expression for the cable extension $T$:

$$T = (J_a - M_{au}M_{uu}^{-1}J_u)^{-1}\left[\left(M_{aa} - M_{au}M_{uu}^{-1}M_{ua}\right)\ddot{X}_a + M_{au}M_{uu}^{-1}s_u - s_a\right] \tag{25}$$

*6.2. Point-to-Point Trajectory Planning Introducing Parameters*

In the process of point-to-point trajectory planning for under-constrained systems, the start and end points of the motion should be chosen as static equilibrium points and the speed at the start and end points should be zero.

For an under-constrained system with $n$ degrees-of-freedom and $m$ cables, there are $m$ controllable degrees of freedom and $n - m$ uncontrollable degrees of freedom. The dynamics Equations (22) and (23) shows that during trajectory planning, only controllable degrees of freedom, $X_a$, can be trajectory planned, while uncontrollable degrees of freedom $X_u$ are influenced by dynamics [20]. When the controllable degree of freedom reaches the end, an uncontrollable degree of freedom cannot be guaranteed to reach the end with zero velocity. Therefore, it is necessary to choose a suitable trajectory plan for the controllable degrees of freedom to ensure that the uncontrollable degrees of freedom reach the end point with zero velocity.

Assume that the trajectory planned for $X_a$ is $x_a(t), t \in [0, T]$, the trajectory of $X_u$ to be found is $x_u(t)$, and the set of differential equations is established by Equation (24) as follows.

$$y = \begin{bmatrix} x_u(t) \\ \dot{x}_u(t) \end{bmatrix}$$

$$\dot{y} = \begin{bmatrix} \dot{x}_u(t) \\ M_{uu}^{-1}\left(s_u + J_u T - M_{ua}\ddot{x}_a\right) \end{bmatrix} = f(y, x_a, \dot{x}_a, \ddot{x}_a) \tag{26}$$

$$y(0) = \begin{bmatrix} x_u(0) \\ 0 \end{bmatrix} := y_0, \quad y(T) = \begin{bmatrix} x_u(T) \\ 0 \end{bmatrix} := y_T$$

where $y_0, y_T$ are the position and velocity of the non-controllable degrees of freedom at the start and end points, respectively, which are both static equilibrium points with zero velocity.

The optimal trajectory for controllable degrees of freedom is achieved when the trajectory $x_a(t)$ is such that Equation (26) holds.

Since Equation (26) has $2 \times (n - m)$ variables and its boundary conditions have $4 \times (n - m)$ constraint. In order to make Equation (26) solvable, $2 \times (n - m)$ parameter $\kappa_1, \cdots, \kappa_{2\lambda}$ needs to be added to the planned trajectory $x_a(t)$.

As an example of a conventional polynomial, a trajectory is planned for controllable degrees of freedom $X_a$ and parameters are introduced for the planned trajectory.

A conventional polynomial point-to-point trajectory is planned as a straight-line path [21] with a trajectory equation of the form:

$$x_a(t) = x_a(0) + (x_a(T) - x_a(0))u(t) \tag{27}$$

where $x_a(0), x_a(T)$ is the position of the controllable degrees of freedom at the start and end points, respectively, $0 \leq u \leq 1$.

For a smoothly derivable point-to-point trajectory $x_a(t)$ of order $r$, the law of motion $u(t)$ is designed as follows:

$$u(t) = \sum_{i=r+1}^{2r+1} a_i \left(\frac{t}{T}\right)^i, \ t \in [0, T] \tag{28}$$

where

$$a_i = \frac{(-1)^{i-r-1}(2r+1)!}{i \cdot r!(i-r-1)!(2r+1-i)!} \tag{29}$$

Introducing parameters to the trajectory of a traditional polynomial programming, the equation of the trajectory is

$$x_a(\kappa, t) = x_a(0) + (x_a(T) - x_a(0))u(\kappa, t) \tag{30}$$

The law-of-motion function $u(\kappa, t)$ can be set to

$$u(\kappa, t) = u(\gamma(\mathbf{\kappa}, t)) = \sum_{i=r+1}^{2r+1} a_i \gamma^i(\mathbf{\kappa}, t) \tag{31}$$

where $a_i = \frac{(-1)^{i-r-1}(2r+1)!}{i \cdot r!(i-r-1)!(2r+1-i)!}$. In order to maintain $u(0) = 0, u(T) = 1, \gamma(\mathbf{\kappa}, t)$. is expressed as follows.

$$\gamma(\mathbf{\kappa}, t) = \alpha t + \sum_{i=2}^{2\lambda+1} \kappa_{i-1} t^i \tag{32}$$

where $\alpha = \frac{1 - \sum\limits_{i=2}^{2\lambda+1} \kappa_{i-1} T^i}{T}$.

Equation (26) is a multivariate marginal differential equation containing parameters that are converted to a multivariate initial differential equation for solution, and the expression of the converted multivariate initial differential equation is as follows.

$$\begin{cases} \dot{y} = f(y, x_a(\kappa, t), \dot{x}_a(\kappa, t), \ddot{x}_a(\kappa, t)) \\ \quad\quad\quad\quad y(0) = y_0 \end{cases} \tag{33}$$

Since the solution of the multivariate initial differential equation removes the constraint $y(T) = y_T$ at the end point, the result can be solved again by setting up the Newton iteration equation $F(\kappa) = y(\kappa, T) - y_T$ to ensure that the end point of the solved trajectory is the same as the planned trajectory.

The trajectory parameters $\kappa_i$ for controlled degree of freedom planning is solved as follows.

1. Set the initial equations of the Newton iterative method: $F(\kappa) = y(\kappa, T) - y_T$, the iterative convergence value: $\zeta$, and the initial values of the parameters to be solved: $\kappa_0$.
2. Solve the multivariate initial differential Equation (33) by taking $t = T$ into the solution $y(\kappa, t)$, and calculating equation $F(\kappa_i)$.
3. If the condition $\|F(\kappa_i)\| \le \zeta$ is satisfied, substitute the coefficient into the trajectory equation $x_a(\kappa, t)$ to obtain the best planning trajectory for the controllable degrees of freedom, and the solution is finished; otherwise, let $\kappa_{i+1} = \kappa_i + J_F^{-1}(\kappa_i)F(\kappa_i)$, and substitute $\kappa_{i+1}$ as the initial value into step 2 to continue the solution.

## 7. Simulation Analysis of Trajectory Planning

### 7.1. 3−6 Under-Constrained Parallel Robot

Point-to-point trajectory planning was performed for the 3−6 (6 degrees of freedom parallel robots with 3 cables) under-constrained parallel robot with the articulation shown in Table 1, and Table 4 shows the end-effector and trajectory planning parameters.

**Table 4.** End-effector and trajectory planning parameters.

| $m(\mathrm{kg})$ | $I_{o\prime}(\mathrm{kg} \cdot \mathrm{m}^2)$ | $X_0$ | $T(\mathrm{s})$ | $X_T$ |
|---|---|---|---|---|
| 0.935 | $\begin{bmatrix} 0.01613 & 0 & 0 \\ 0 & 0.01599 & 0 \\ 0 & 0 & 0.03211 \end{bmatrix}$ | $\begin{bmatrix} 0 \\ 0.59 \\ 1 \\ -0.43 \\ 0 \\ 0 \end{bmatrix}$ | 1.5 | $\begin{bmatrix} -0.15 \\ 0.8 \\ 1.17 \\ -0.05 \\ 0.26 \\ 0 \end{bmatrix}$ |

In Table 4, $m$ is the mass of the end-effector and $I_{o\prime}$ is the inertia tensor of the end-effector in a local coordinate system where the origin of the local coordinate system coincides with the center of mass. $X_0$, $X_T$ is the initial and end pose of the trajectory planning and both are static equilibrium points. Figure 5 shows the initial state and end state for unconstrained parallel robot trajectory planning.

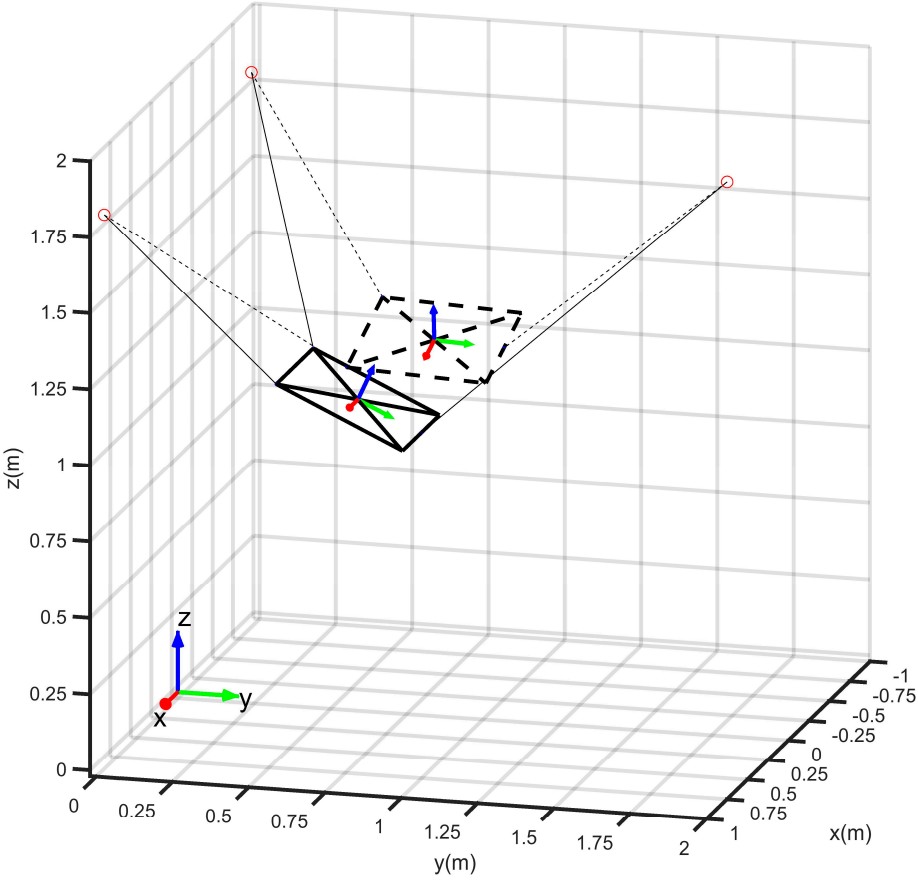

**Figure 5.** Initial state (solid line) and end state (dashed line) for unconstrained parallel robot trajectory planning.

The controllable degrees of freedom are $X_a = \begin{bmatrix} ^O x_{O\prime}, ^O y_{O\prime}, ^O z_{O\prime} \end{bmatrix}$ and the uncontrollable degrees of freedom are $X_b = [\alpha, \beta, \gamma]$. These are the angles of rotation around the $x$-axis, $y$-axis and $z$-axis respectively.

For the traditional polynomial trajectory planning method, $r = 3$ is taken to ensure that the trajectory is smoothly derivable to the third order, and $a_i$ is calculated via Equation (29) with the following results.

$$a_i = \begin{bmatrix} a_4 \\ a_5 \\ a_6 \\ a_7 \end{bmatrix} = \begin{bmatrix} 35 \\ -84 \\ 70 \\ -20 \end{bmatrix}$$

Substitute $a_i$ into Equation (27) to find the conventional polynomial trajectory $x_a(t)$, and solve Equation (33) for the multivariate initial value differential equation.

For the polynomial trajectory planning method with the introduction of parameters, take $r = 3$, whose coefficients $a_i$ are the same as those of a conventional polynomial trajectory. Let the initial value of the parameter vector $\kappa$ be the 6$-$dimensional zero vector $\zeta = 1 \times 10^{-8}$, and solve for the parameters using Newton's iterative method to obtain the following results.

$$\kappa = \begin{bmatrix} \kappa_1 \\ \kappa_2 \\ \kappa_3 \\ \kappa_4 \\ \kappa_5 \\ \kappa_6 \end{bmatrix} = \begin{bmatrix} -5.52007049 \\ 15.73380808 \\ -24.49999798 \\ 21.38293401 \\ -9.84359244 \\ 1.85786785 \end{bmatrix}$$

Substituting the parameter $\kappa$ into Equation (30), the traditional polynomial trajectory $x_a(\kappa, t)$ is obtained and solved for the multivariate initial differential Equation (33).

The solution results of the traditional polynomial trajectory planning method are compared with those of the polynomial trajectory planning method with parameters, and the comparison results are shown below.

Figure 6 shows the pose of the end-effector in the trajectory. In this case, the direction drops from 0 m to $-0.15$ m, the direction increases from 0.59 to 0.8 and the direction increases from 1 m to 1.17 m. The orientation angle also changes under the influence of the system dynamics, where $\alpha$ rises from $-0.43$ rad to $-0.05$ rad and $\beta$ from 0 rad to 0.26 rad (rotated about $z$-axis) and $\gamma$ remains largely stable during the process. The trajectory start and end points of both planning algorithms are consistent with $X_0$, $X_T$ in Table 4 in terms of the pose curves.

Figure 7 shows the velocity curves of the trajectories solved by the two planning algorithms. It can be seen that the difference in velocity between the trajectories planned by the two algorithms is more obvious. Both planning algorithms achieve the boundary condition of zeroing the velocity at the start and end point in the direction of $x, y, z$. However, for the angular velocity profile of $\alpha, \beta, \gamma$, the traditional trajectory planning algorithm cannot guarantee stationary at the end point, and the trajectory does not zero at $\alpha, \beta, \gamma$ of 2.5 s, which cannot guarantee the trajectory boundary condition.

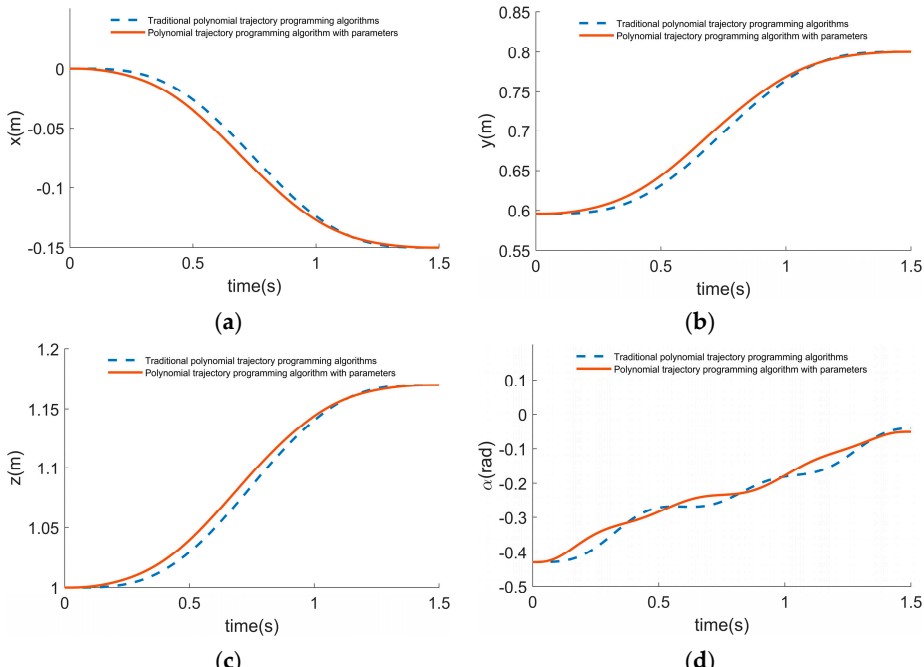

**Figure 6.** *Cont.*

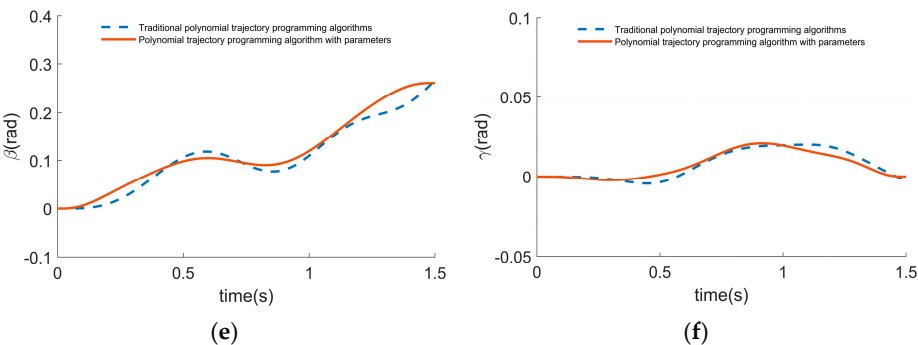

**Figure 6.** Trajectory poses solved under two planning algorithms. (**a**) Comparison of *x*-direction displacement; (**b**) Comparison of *y*-direction displacement; (**c**) Comparison of *z*-direction displacement; (**d**) Comparison of $\alpha$ angular; (**e**) Comparison of $\beta$ angular; (**f**) Comparison of $\gamma$ angular.

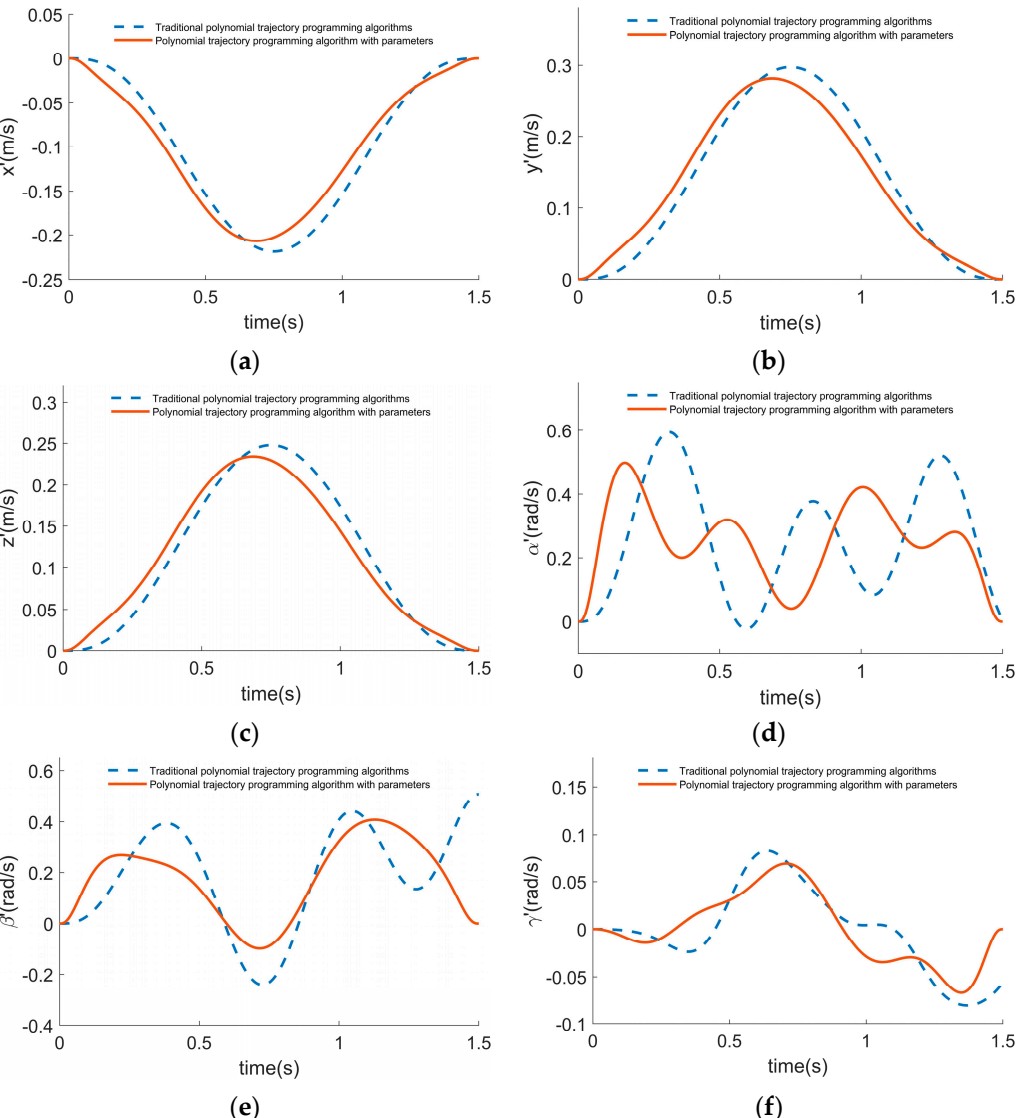

**Figure 7.** Velocity solved under two planning algorithms. (**a**) Comparison of $x-$direction velocity; (**b**) Comparison of $y-$direction velocity; (**c**) Comparison of $z-$direction velocity; (**d**) Comparison of $\alpha-$angle velocity; (**e**) Comparison of $\beta-$angle velocity; (**f**) Comparison of $\gamma-$angle velocity.

### 7.2. 0−0−2 Under-Constrained Planar Hybrid Robot

Point-to-point trajectory planning for an under-constrained cable-driven robot with 0-2-2 (Figure 8). The center of mass of each link is at its geometric center. The coordinates of the articulation points and the link parameters for the cable-driven robot are shown in Tables 2 and 3, while the following Table 5 shows the link inertia and trajectory planning related parameters.

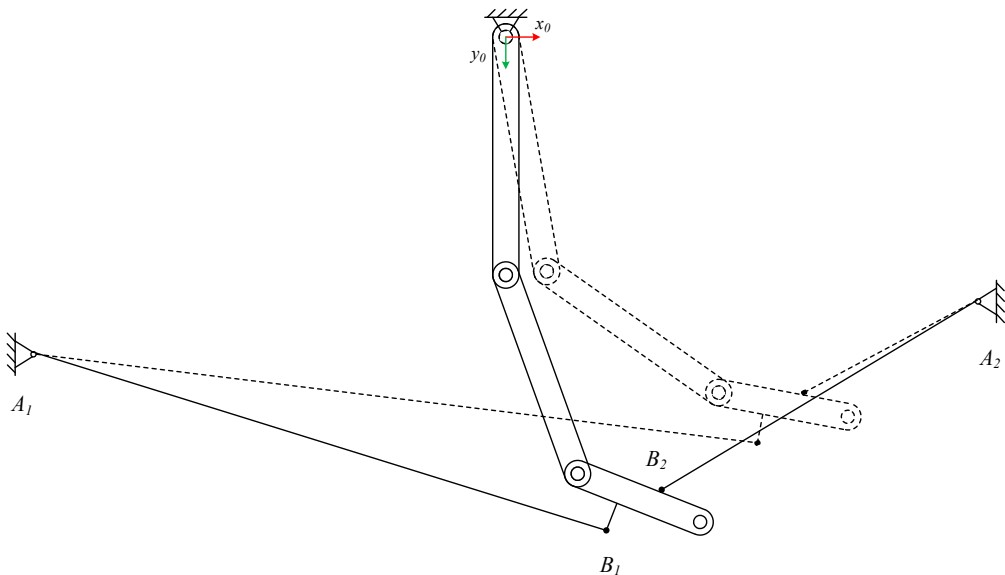

**Figure 8.** Initial state (solid line) and end state (dashed) for under-constrained cable-driven robot in trajectory planning.

**Table 5.** Parameters about Figure 8 to link rotational inertia and trajectory planning.

| $I_{zi}(\mathrm{kg \cdot m^2})$ | $X_0(\mathrm{rad})$ | $T(\mathrm{s})$ | $X_T(\mathrm{rad})$ |
|---|---|---|---|
| $I_{z1} = 0.2$ $I_{z2} = 0.06$ $I_{z3} = 0.006$ | $\begin{bmatrix} 90 \\ -20 \\ -48.61 \end{bmatrix} \times \frac{\pi}{180}$ | 1 | $\begin{bmatrix} 80 \\ -45 \\ -24.44 \end{bmatrix} \times \frac{\pi}{180}$ |

In Table 5, $I_{zi}$ is the mass product of inertia of the $i$ link around the $z$-axis. $X_0$, $X_T$ are the starting and ending points of the trajectory plan, and both are static equilibrium points.

The controllable degrees of freedom are $X_a = [\theta_1, \theta_2]$, and the uncontrollable degrees of freedom are $X_b = [\theta_3]$.

For the traditional polynomial trajectory planning method, $r = 3$ is taken to ensure that the trajectory is smoothly derivable to the third order, and $a_i$ is calculated via Equation (29) with the following results.

$$a_i = \begin{bmatrix} a_4 \\ a_5 \\ a_6 \\ a_7 \end{bmatrix} = \begin{bmatrix} 35 \\ -84 \\ 70 \\ -20 \end{bmatrix}$$

Substitute $a_i$ into Equation (27) to obtain the traditional polynomial trajectory $x_a(t)$, and solve Equation (33) for the multivariate initial differential equation.

For the polynomial trajectory planning method with the introduction of parameters, take $r = 3$, whose coefficients $a_i$ are the same as those of a conventional polynomial trajectory. Let the initial value of the parameter vector $\kappa$ be a 2−dimensional zero vector, $\zeta = 1 \times 10^{-8}$, and solve for the parameters using Newton's iterative method to obtain the following results.

$$\kappa = \begin{bmatrix} \kappa_1 \\ \kappa_2 \end{bmatrix} = \begin{bmatrix} 0.58865332 \\ -0.29981383 \end{bmatrix}$$

Substituting the parameter $\kappa$ into Equation (30), the conventional polynomial trajectory $x_a(\kappa, t)$ is obtained and solved for the multivariate initial differential Equation (33).

A comparison of the solution results of the traditional polynomial trajectory planning method with those of the polynomial trajectory planning method with the introduction of parameters is shown below.

The joint angle curves for the two planning algorithms are shown in Figure 9. The joint angle $\theta_1, \theta_2$ starts and ends at the same point in both planning algorithms, while angle $\theta_1$ decreases from 90° to 80° and angle $\theta_2$ changes from $-25°$ to $-45°$. As can be seen from Figure 7c, in the polynomial trajectory planning method with the introduction of parameters, the start and end points of the $\theta_3$ curve are the same as $X_0, X_T$ in Table 5, whereas the traditional polynomial trajectory planning method does not guarantee this.

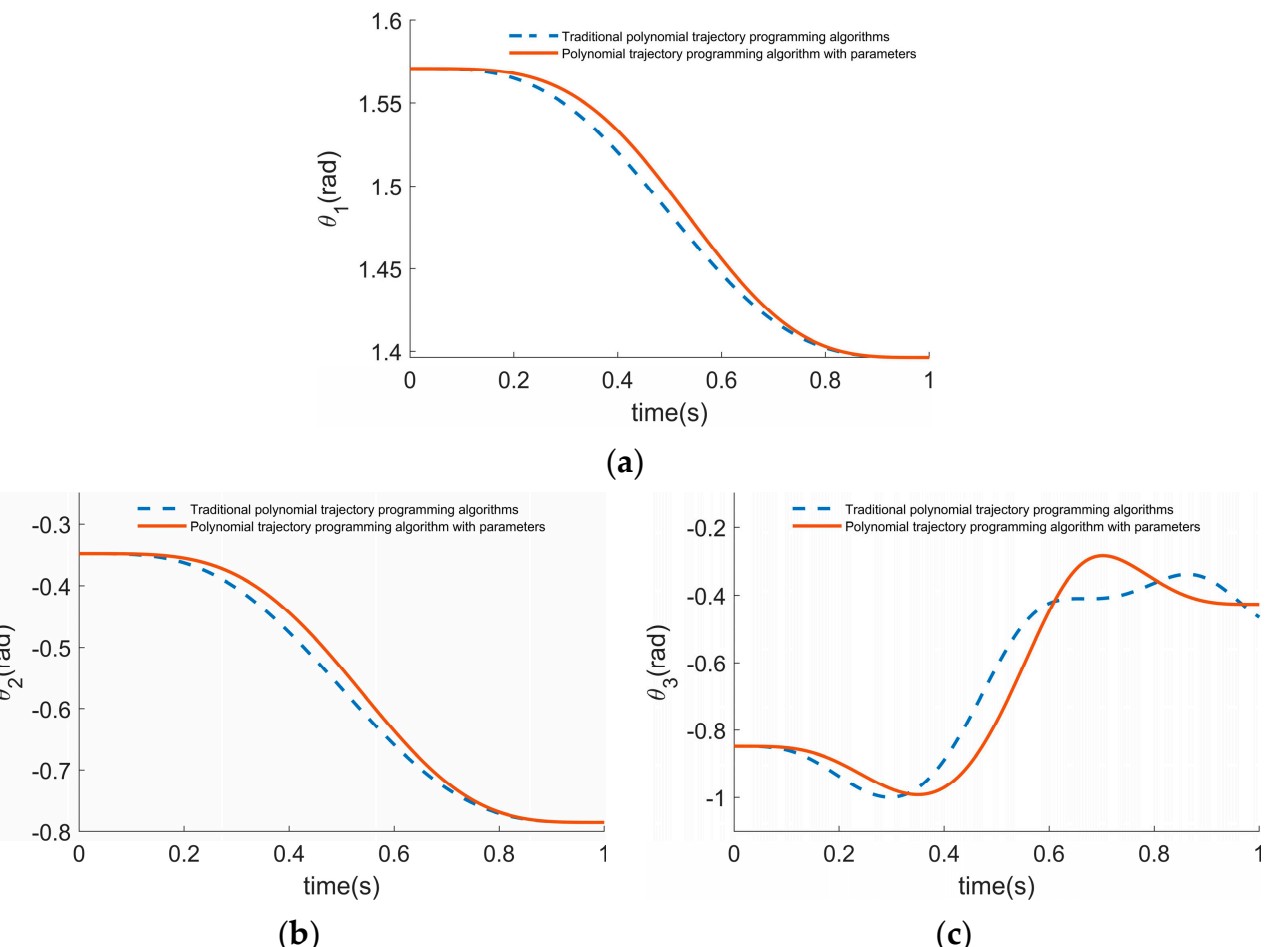

**Figure 9.** Trajectory joint angles solved under the two planning algorithms. (**a**) Comparison of $\theta_1-$angle; (**b**) Comparison of $\theta_2-$angle; (**c**) Comparison of $\theta_3-$angle.

The angular velocity of the two planning algorithms is shown in Figure 10. Both planning algorithms reach the boundary condition of zero at the start and end points for the angular velocity of $\theta_1, \theta_2$. As can be seen in Figure 10c, for the angular velocity profile of $\theta_3$, stationarity is not guaranteed at the end point using the conventional trajectory planning algorithm, whereas the polynomial trajectory planning method with the introduction of parameters maintains stationarity at the end point.

In summary, for under-constrained parallel or planar hybrid systems, the traditional polynomial trajectory planning algorithm cannot guarantee zero velocity at the end of the trajectory. In contrast, the polynomial trajectory planning algorithm with the introduction of parameters at the start and end points keeps the under-constrained system stable without oscillation and satisfies the trajectory planning requirements.

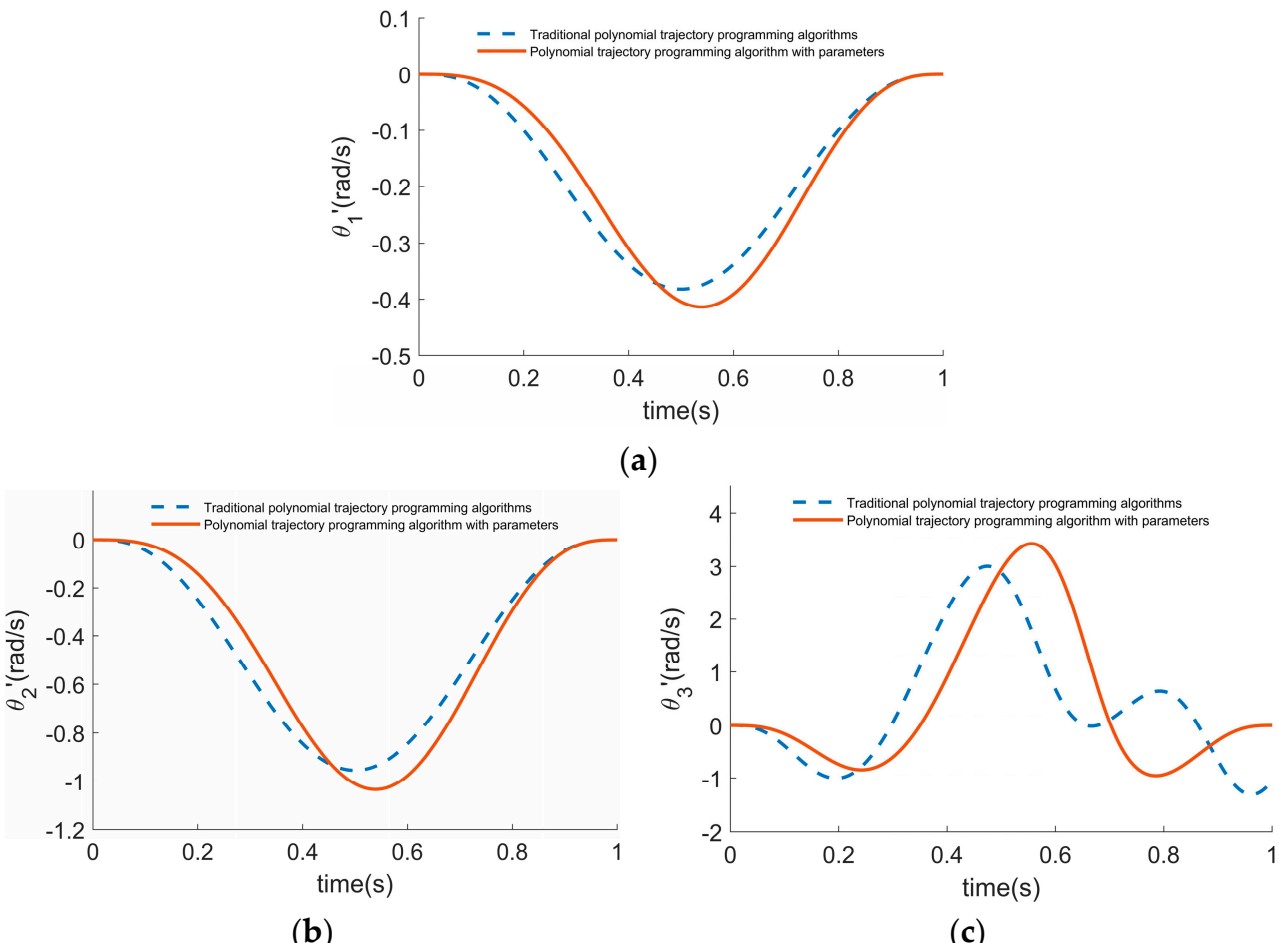

**Figure 10.** Angular velocity of the solution under the two planning algorithms. (**a**) Comparison of $\theta_1-$angular velocity; (**b**) Comparison of $\theta_2-$angle velocity; (**c**) Comparison of $\theta_3-$angle velocity.

## 8. Conclusions

In this paper, the force closure equations and geometric closure equations for incompletely constrained cable traction systems (parallel and planar hybrid) are developed, the equations are solved jointly using a convex optimization solution method with boundary conditions, a static equilibrium inverse kinematic model of the incompletely constrained cable traction system is developed, and the static equilibrium workspace is solved.

Point-to-point trajectory planning algorithms for incompletely constrained cable traction systems are investigated. Traditional purely algebraic methods such as polynomial functions or trigonometric functions for planning the trajectory of the end-effector do not take into account the dynamical model of the system, which can cause oscillations of the system at the start and end points due to insufficient controllable degrees of freedom on the UCR. In this paper, a dynamical model of the UCR is developed, which consider the cable force, external force, controllable position and orientation and uncontrollable position and orientation of the end-effector, etc. Based on this model, the traditional polynomial-based point-to-point trajectory planning algorithm is improved by adding $2 \times (n-m)$ parameters to the kinematic law function $u(t)$. The constraints of the dynamics model are incorporated into the trajectory planning process to achieve point-to-point trajectory planning for the UCR. The results show that the trajectory of the improved algorithm is smooth and derivable, and the end-effector is stationary and stable without oscillation at the start and end points, which proves the effectiveness of the optimized algorithm.

**Author Contributions:** Conceptualization, Q.D., Q.Z. and T.W.; methodology, Q.D.; software, Q.Z.; validation, Q.D. and Q.Z.; writing—original draft preparation, Q.Z.; writing—review and editing, Q.D. and Q.Z. All authors have read and agreed to the published version of the manuscript.

**Funding:** This research was funded by Key Research and Development Program of Shaanxi (Program No. 2022GY-314).

**Institutional Review Board Statement:** Not applicable.

**Informed Consent Statement:** Not applicable.

**Data Availability Statement:** Not applicable.

**Conflicts of Interest:** The authors declare no conflict of interest. The funders had no role in the design of the study; in the collection, analyses, or interpretation of data; in the writing of the manuscript; or in the decision to publish the results.

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
