# Peer review of "Consistent Solution Strategy for Static Equilibrium Workspace and Trajectory Planning of Under-Constrained Cable-Driven Parallel and Planar Hybrid Robots"

_machines, doi:10.3390/machines10100920_

Round 1
Reviewer 1 Report
The paper focuses on computing the kinematic and dynamic relations of two cable-driven robots. Using the kinematic equation the authors calculate a range of robot positions, either in cartesian space for the parallel robot or in joint space for the serial one. Having the range of motion, the authors validate each point as a static equilibrium point, resulting in a validated workspace where each robot can be placed in static equilibrium.
The paper has a lot of English mistakes. I have found many typos and phrases that are incorrect and I might have missed others:
- Typos on lines: 29, 52, 187.
- Sentences that require rephrasing on lines: 37, 41-42, 188-189, 224, 237.
The used references are old and focused on very few points from the introduction section.
Figure 2 is not very explicit. It shows the planar cable-driven serial robot. As stated in the document, the notation for the cable position vector in the robot bar frames is Bij, but in figure 2, it is written with Pij. Please correct either the figure or the notation in the text. Moreover, the authors can add the Cartesian coordinates used and even the center of mass for each link with the gravity direction, since this is the force required for the static equilibrium.
Overall, the paper requires a high amount of changes before publishing. The computation of a robot workspace with constraints is not enough to publish in this journal. The authors should add a use case that requires these computations, control methods, and advanced algorithms.
After major changes, I recommend resubmitting the paper.
Reviewer 2 Report
The article presents a pending approach, i.e. a consistent solution strategy for static equilibrium workspaces of various types of robots with limited constraints. This approach is based on the use of the least squares error rate to find all solutions of a rope drive robot with constraints.
In my opinion, the topic is interesting and quite innovative. A better idea would be to compare the proposed methods with another method, such as the PSO method.
The results presented by the authors are quite convincing and confirm the legitimacy of using this method, but in my opinion there are no drawings and illustrations that would allow the reader to read the article. The article provides a limited example to help the reader understand the method used. However, I have comments to the authors:
The literature review is very poor, I propose to expand it further
I propose a broader development of conclusions,
I propose to add more references as there are many articles on this topic.
Round 2
Reviewer 1 Report
Dear authors the paper quality has significantly improved by adding new sections. Thank you for understanding that publishing high-quality journal papers requires much more mathematical analysis and results.
I now have only a few remarks on some errors in writing the paper. These are as follows:
1. Line 78, please correct the sentence: "Based on this, a Consistent Solution Strategies Consistent Solution Strategies of Point-To-78 Point Trajectory Planning Introducing Parameters".
2. Line 280, X" seems out of place, probably it should not be there. Please correct, or rephrase.
3. Line 295-296, "The kinetic Equation": please correct, the authors are referring to the dynamic equations 22 and 23. A kinetic equation is an equation for non-equilibrium particles.
4. Images from Figures 6, 7, 9, and 10 are hard to read, please update them with high-quality ones.
5. Line 382, please correct: "the direction drops from 0m to 0.15m", it should be -0.15m.
Author Response
The author’s answer:
- We feel sorry for our carelessness. In our resubmitted manuscript, for question 1,2,3, and 5, the typo is revised. Thanks for your correction. And here we did not list the changes but marked in yellow in the revised paper.
- For question 4, we have changed Figures 6, 7, 9, and 10 to high-quality ones.
Reviewer 2 Report
The authors more or less responded to my comments, I think you can take over the article.
Author Response
Thank you very much for your comments and professional advice. These opinions help to improve the academic rigor of our article. Based on your suggestion and request, we have made corrected modifications to the revised manuscripts. We hope that our work can be improved again. Furthermore, we would like to show the details as follows:
The author’s answer:
- We feel sorry for our carelessness. In our resubmitted manuscript, for question 1,2,3, and 5, the typo is revised. Thanks for your correction. And here we did not list the changes but marked in yellow in the revised paper.
- For question 4, we have changed Figures 6, 7, 9, and 10 to high-quality ones.